# Variational Mode Decomposition Based Time-Varying Force Identification of Stay Cables

Shitong Hou [1,2], Bin Dong [1,2], Jianhua Fan [1,2], Gang Wu [1,2,]*, Haochen Wang [1,2], Yitian Han [1,2] and Xiaojin Zhao [3]

1   Key Laboratory of Concrete and Prestressed Concrete Structures of the Ministry of Education, Southeast University, Nanjing 210096, China; sthou@seu.edu.cn (S.H.); bdonghust@gmail.com (B.D.); fjhseu@foxmail.com (J.F.); wanghaochen1995@163.com (H.W.); skye_hanyt@seu.edu.cn (Y.H.)
2   National and Local Joint Engineering Research Center for Intelligent Construction and Maintenance, Southeast University, Nanjing 210096, China
3   Key Laboratory of Road Construction and Maintenance Technology and Transportation Industry in Loess Area, Shanxi Communications Technology R&D Co. Ltd., Taiyuan 030027, China; agfdes@126.com
*   Correspondence: g.wu@seu.edu.cn; Tel.: +86-025-5209-1304

**Featured Application: Intelligent Detection of Bridges.**

**Abstract:** Stay cables are important structural members of cable-stayed bridges, which play a significant role in the health monitoring and assessment of cable-stayed bridges. The in-service cable force, which varies from the effects of vehicle load, wind load and other environmental factors, may cause fatigue damage in stay cables. Traditional force identification methods can only calculate the time-average cable force instead of the instantaneous force. A novel method has been proposed in this paper for identifying time-varying cable tension based on the variational mode decomposition (VMD) method. This recent method decomposes signals and adaptively estimates instantaneous frequency combined with the Hilbert–Huang transform method. In the proposed study, the time-varying modal frequencies were identified from stay cable acceleration data, and then the time-varying cable tension was identified by the relationship between cable tension and identified fundamental frequency. Scaled and full-scale models of stay cables were implemented successively to illustrate the validity of the proposed method. The results showed that the variational mode decomposition (VMD) method has a good effect on identifying the time-varying cable forces, even the sudden changes in cable force. According to the cable force identification results, the maximum error was 8.4%, which meets the actual application of time-varying cable force measurements. An on-site test was also implemented to monitor the cable force during a construction period, and the results showed that the proposed method can provide accurate real-time results for evaluation and decision-making.

**Keywords:** variational mode decomposition; stay cables; time-varying force; cable force identification; hilbert–huang transform

## 1. Introduction

Stay cables are the main load-bearing and force-transmitting members of cable-stayed bridges. Dynamic adjustment of cable force can adjust the overall symmetry, stability, and resistance to the structural deformation in a balanced and stable state. During the service process of cable-stayed bridges, stay cables bear their own dead load and external live load for a long time, and the cable force changes with the external load, which tends to induce accumulated fatigue damage and affects the service life of stay cables. Some stay cables become severely damaged and need to be replaced before they reach the designed service life, which seriously affects the use of cable-stayed bridges. Therefore, research on how to identify the time-varying cable force of stay cables is of great importance, which can not only prepare for further stay cable fatigue analysis but also provide a scientific basis for the maintenance of stay cables and cable-stayed bridges.

Many studies on the methods of cable force identification have been carried out [1–5]. Currently, cable force identification methods mainly include the static force-based method, which directly measures the cable force of a stay cable by embedding a tension sensor at the end of the stay cable's anchoring; the vibration-based method [1,6,7], which calculates cable force by the cable force-fundamental frequency formula; the method in which vibration frequency is calculated by accelerations of the stay cable using fast Fourier transform; and the magnetic flux-based method [8,9], which uses the correspondence between magnetic flux and cable force. For any type of stay cable method, several sets of tests under stress and temperature have been performed in the laboratory to establish the relationship between magnetic flux, structural stress and temperature, which can then be used to identify the cable force of a specific material. For existing bridges, it is not possible to install a tension sensor during the service period, and it is cost-consuming to use the magnetic flux-based method. In recent years, the vibration-based method has been widely studied, which has the advantages of inexpensiveness and convenience.

Currently, the calculation models of cable forces based on vibration [10] are mainly divided into four categories: tense string model theory, simple-supported beam model theory, fixed-supported beam theory, and complex boundary model theory. The tension string theory equates the stay cable to a tensionless string without stiffness and sag, and a vibration differential equation is established to calculate the cable force:

$$F - 4mL^2 \left( \frac{f_n}{n} \right)^2 = 0 \tag{1}$$

where $F$ is the cable force, $L$ is the length of the stay cable, $f_n$ is the $n$th vibration frequency of the stay cable, $n$ is the order of the vibration mode, and $m$ is the mass per unit cable length. Therefore, as long as the vibration frequency of the stay cable is obtained, the cable force value can be calculated. However, in practical applications, stay cables have a certain stiffness and sag. Bridge experiments [11,12] which ignored bending stiffness were implemented to identify the cable force according to the tension string theory, and the error between the measured value of the cable force and true value remained within 5%. The simple-supported beam model equates the stay cable to a horizontal beam that hinges at one end and is simply supported at another end. Stay cables usually have stiffness, so they can be considered as simply supported beams. According to this theory, the cable force can be calculated according to the following formula:

$$F - 4mL^2 \left( \frac{f_n}{n} \right)^2 + EI \frac{n^2 \pi^2}{L^2} = 0 \tag{2}$$

where $EI$ refers to the bending stiffness of the stay cable, and the meanings of other symbols are the same as in Equation (1). Related studies [13,14] proposed several practical simplified formulas according to this theory and the measured natural frequency of the stay cable. The measured natural frequency according to their theory was only 3% different from the theoretical value. Compared with the simple-supported beam model, the fixed-supported beam model uses fixed-support constraints at both ends. Specifically, the boundary conditions are different. Due to the clamped constraint, the displacement and rotation angle at both ends is 0. The cable force calculation formula is shown in Equations (3)–(6):

$$2\sigma\epsilon[1 - \cos(\delta l)\cos(\epsilon l)] + \left( \epsilon^2 - \delta^2 \right) \sin(\delta l)\sinh(\epsilon l) = 0 \tag{3}$$

$$\delta l = \frac{\xi}{\sqrt{2}} \sqrt{\sqrt{1 + \left( \frac{2\alpha_1^2}{\xi} \varphi_n \right)^2} - 1} \tag{4}$$

$$\epsilon l = \frac{\xi}{\sqrt{2}} \sqrt{\sqrt{1 + \left(\frac{2\alpha_1^2}{\xi} \varphi_n\right)^2} + 1} \tag{5}$$

$$\xi = \frac{F}{EI}l \qquad \varphi_n = \frac{f}{\frac{\alpha_1}{2\pi l^2}\sqrt{\frac{EI}{m}}} \tag{6}$$

where $l$ is the length of the stay cable, $\varphi_n$ and $F$ can be iteratively calculated according to the given $\alpha_1 = 4.730$ and initial value $\xi$. The meanings of other symbols are the same as in Equation (1). In practical applications, the boundary conditions of stay cables are more complicated [15,16] and may be between simple-supported and fixed-supported. Related studies [17,18] established a unified and simplified model of stay cables under elastic boundary conditions. Unlike the previous three models, the elastic boundary can withstand a certain bending moment and a certain angle by which the bending moment can be determined. The calculation formula is shown in Equation (7):

$$F - \frac{4m\pi^2 l^2 f_n^2}{(n\pi - \theta_n)^2 \left(1 + \beta^2 (n\pi - \theta_n)^2\right)} = 0 \tag{7}$$

where $l$ is the length of the stay cable, and $\theta_n$ and $\beta$ are parameters related to boundary conditions and the meanings of other symbols are the same as in Equation (1).

The above theories and vibration models provide formulas for vibration-based cable force measurement methods, but there are still few methods for measuring the time-varying cable force. Li et al. [19] used the Kalman filter algorithm to estimate the time-varying cable force according to the measured acceleration data with extra wind speed data measured on site. Yang et al. [20] proposed an algorithm called complexity pursuit to identify time-varying cable tension forces according to acceleration data. This method requires two independent acceleration data points from two accelerometers on a cable. Bao et al. [21] proposed a sparse time-frequency decomposition method [22–24] that calculates the time-varying cable force by estimating the instantaneous frequency.

Aiming at time-varying cable force identification, this present study leverages the advantages of variational mode decomposition (VMD) and the Hilbert–Huang transformation (HHT) theories to propose a satisfactory method, and a single accelerometer without extra information such as wind speed was used for more convenient application. In addition, the proposed method shows better performance in cases where the cable force mutation amplitude is more than 15% compared with existing methods, including the sparse time-frequency decomposition method. Scaled and full-scale experiments were both conducted for the comparison between calculated and real cable forces. Several conditions were designed to simulate the time-variance and sudden change in cable forces. In addition, an on-site bridge was also implemented to monitor the cable force during a maintenance period. This paper is organized as follows. Section 2 illustrates the proposed method using VMD and HHT for cable force identification. Section 3 describes the experimental results under different conditions in scale and full-scale tests. Section 4 introduces the on-site test of cable force monitoring. Finally, Section 5 draws a conclusion about the feasibility of the proposed method.

## 2. Variational Mode Decomposition Analysis

The vibration data of a stay cable often contain modal information of each order, that is, components with different frequencies. In the signal processing process, the modal decomposition method [25] is generally used to decompose the original signal into vibration signals of various orders for analysis. Variational mode decomposition [26] (VMD) is a new modal decomposition method that searches for the optimal solution of a constrained variational model established by a specified center frequency to achieve adaptive signal decomposition. For each intrinsic mode function [27–29] (IMF), the frequency center and

bandwidth of the component are continuously and iteratively updated to complete the final adaptive decomposition of the signal in the frequency domain. The IMF component is overwhelmed by noise, and the signal characteristic component is decomposed.

VMD algorithm is a completely nonrecursive signal decomposition method that can determine the center frequency and bandwidth of each intrinsic mode function (IMF) by iteratively searching for the optimal solution of the variational model and then adaptively decomposing the original signal into the sum of the individual IMF components. The signal is decomposed into a series of IMFs by VMD processing, and each IMF can be represented as an AM-FM [30] signal $u_k(t)$ as Equation (8) shows:

$$u_k(t) = A_k(t) \cos(\phi_k(t)) \tag{8}$$

where $A_k(t)$ is the instantaneous amplitude and $\phi_k(t)$ is the instantaneous phase. The instantaneous frequency is defined as $\omega_k(t) = \phi_k'(t)$. The changes in $\phi_k(t)$ and $A_k(t)$ are relatively slow to $\omega_k(t)$. Within the time interval $[t - \delta, t + \delta]$, $u_k(t)$ can be regarded as the harmonic signal with amplitude $A_k(t)$ and frequency $\omega_k(t)$. Three steps are needed to estimate each IMF: (1) calculating the analytical signal associated with each mode by the Hilbert transform, (2) adjusting the center frequency of each estimate by adding an index term and transforming the spectrum into the baseband, and (3) smoothing the demodulated signal and estimated bandwidth by the H1 Gauss method. Assuming that the signal is decomposed into *K* IMFs after VMD processing, the variational constraint model is shown in Equation (9):

$$\min_{\{\mu_k\},\{\omega_k\}} \left\{ \sum_{k=1}^{K} ||\partial_t \left[ \delta(t) + \frac{j}{\pi t} \otimes \mu_k(t) \right] e^{-j\omega_k(t)}||_2^2 \right\}$$

$$s.t \quad \sum_{k=1}^{K} \mu_k(t) = f \tag{9}$$

where $\mu_k(t)$ is the *k*th signal component, $\omega_k(t)$ represents the center frequency of each component, which can be characterized as the instantaneous frequency of each order vibration mode of the stay cable, $\partial_t$ refers to the derivative of time, $\delta(t)$ represents the unit pulse function, $\otimes$ is the convolution operation, and *j* is the plural sign. The above problem is actually a convex optimization problem. To solve the above optimization problem, the VMD algorithm introduces $L_2$ regularization constraints and Lagrangian multipliers and turns Equation (9) into the augmented Lagrangian function as Equation (10) follows:

$$L(\{\mu_k\}, \{\mu_k\}, \lambda) = \alpha \sum_{k=1}^{K} ||\partial_t \left[ \delta(t) + \frac{j}{\pi t} \otimes \mu_k(t) \right] e^{-j\omega_k(t)}||_2^2$$

$$+ ||f(t) - \sum_{k=1}^{K} \mu_k(t)||_2^2 \tag{10}$$

$$+ < \lambda(t), f(t) - \sum_{k=1}^{K} \mu_k(t) >$$

where $f(t)$ is the original signal, $\lambda(t)$ is the Lagrange function multiplier, $\alpha$ is the bandwidth parameter, and the meanings of the remaining symbols are the same as in Equation (9).

For the above equation-constrained optimal solution of the multivariate function, an alternating direction multiplier algorithm (ADMM) can be used to solve iteratively, and the various order components of the signal can be obtained. The next step is to find the instantaneous frequency of each eigenmode component. Instantaneous frequency is an important physical quantity that characterizes the transient frequency characteristics of a signal at a local time point. For most signals, especially natural signals, the frequency

generally changes with time. The instantaneous frequency is defined as the derivative of phase versus time, as shown in Equation (11):

$$\omega(t) = \dot{\varphi}(t) = \frac{\mathrm{d}[\tilde{f}(t)/f(t)]}{\mathrm{d}t} \tag{11}$$

where $\varphi(t)$ is the instantaneous phase and $\tilde{f}(t)$ is the signal obtained from the Hilbert–Huang transform and can be calculated as Equation (12):

$$\tilde{f}(t) = \frac{1}{\pi t} \otimes f(t) = \frac{1}{\pi}\int_{-\infty}^{+\infty}\frac{f(\tau)}{t-\tau}\mathrm{d}\tau \tag{12}$$

where $\otimes$ is the convolution operation. The obtained analytical signal $Z(t)$ is calculated as Equation (13) and Equation (14):

$$Z(t) = f(t) + i\tilde{f}(t) = a(t)e^{i\theta(t)} \tag{13}$$

$$a(t) = [f(t)^2 + \tilde{f}(t)^2]^{\frac{1}{2}} \quad \theta(t) = \arctan(\frac{\tilde{f}(t)}{f(t)}) \tag{14}$$

Furthermore, the instantaneous frequency can be obtained by deriving the instantaneous phase as Equation (15) shows:

$$\omega(t) = \frac{1}{2\pi}\frac{\mathrm{d}\theta(t)}{\mathrm{d}t} \tag{15}$$

The Hilbert transform can adaptively generate the "base", which is the IMF component after the VMD algorithm is decomposed. Other signal decomposition methods include the Fourier transform, whose "base" is a trigonometric function, and the wavelet transform , whose "base" is a wavelet base that satisfies the tolerance condition, which is also selected in advance. In practice, it is difficult to choose the appropriate "base", so the Hilbert transform is relatively more suitable. In addition, the Fourier transform, short-time Fourier transform, and wavelet transform are all constrained by Heisenberg's uncertainty principle. The product of the time window and the frequency window must be a constant. The Hilbert transform is not subject to this criterion, so it is also applicable to abrupt signals. After the instantaneous frequency of the cable vibration acceleration signal is obtained, the time-varying cable force can be calculated according to the Equation (16):

$$F = 4mL^2\omega_1^2 - EI\frac{\pi^2}{L^2} \tag{16}$$

where $\omega_1$ is the first-order instantaneous frequency of the stay cable.

## 3. Stay Cable Experiments

To verify the feasibility of the proposed method, scaled and full scale stay cable models were both conducted in this paper. Due to the large actual size of the cable, the scale-model experiment was easier to implement and the performance of proposed methods can be validated. Then, a actual cable was selected and the cable force identified results derived from the proposed method were studied from this full-scale model experiment.

### 3.1. Scale-Model Experiment

To verify the feasibility of the proposed method, a scaled stay cable model was established in this paper first by using a single strand of steel wire to simulate the vibration of the cable under different tensions and obtained vibration acceleration information. The scaled model is shown in Figure 1. The distance between the anchoring ends of the steel wire was 15 m, the mass per unit length was 0.25 kg/m, and the diameter was 8 mm, the boundary condition of this experiment is that one end is fixed and the other is simply-

supported, the left part in this Figure is the enlarged sketch of tension sensor and anchor. Only the acceleration information is necessary in the proposed mothod, which can be obtained by only one acceletation sensor. However, to prevent cases in which the sensor data were abnormal, an additional sensor was added as a backup. Two acceleration sensors were fixed at 3.8 m and 4.8 m from the anchor point on the side of the tension sensor, the sampling frequency of which was 200 Hz, and the stay cable was excited by a blower with 550 kW motor power; the field layout of the entire experiment is shown in Figure 2. Considering the tensile bearing capacity of the cable, the range of the tension force in this cable was from 5 kN to 7.5 kN. In addition, to make the cable vibration more obvious, a baffle was made in front of the blower to enlarge the wind affected region. A total of four operating conditions were set in this experiment, as shown in Table 1, where operating conditions 1 and 2 were under constant tensile force, and operating conditions 3 and 4 were under sudden changing tensile force.

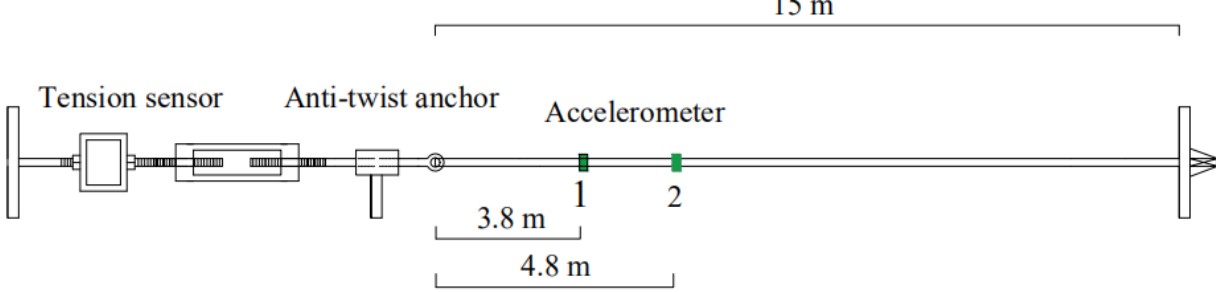

**Figure 1.** Design drawing of the stay cable in the scale model experiment.

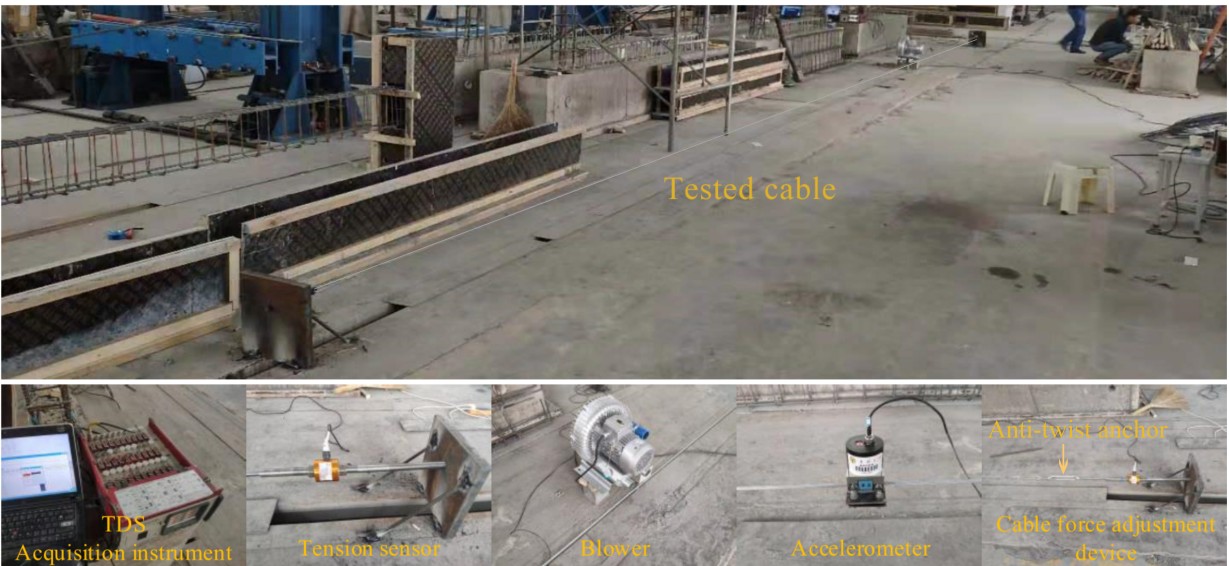

**Figure 2.** Field layout of the stay cable in the scale model experiment.

**Table 1.** Information of designed conditions in the scale model experiment.

| Conditions | Description |
| :---: | :---: |
| 1 | F = 6.0 kN in t ∈ [0 s, 102 s] |
| 2 | F = 5.0 kN in t ∈ [0 s, 102 s] |
| 3 | F = 6.0 kN in t ∈ [0 s, 60 s] F = 5.3 kN in t ∈ [60 s, 108 s] |
| 4 | F = 5.0 kN in t ∈ [0 s, 60 s] F = 7.3 kN in t ∈ [60 s, 108 s] |

Irregular random interference signals usually occur in the collected vibration signals, which causes that the deviation of the sampling values of the signal points is too large, and the collected signals may generate a large number of "burrs". To reduce the influence of the interference signal on the collected signal, the original signal needs to be smoothed before the signal is analyzed according to Equation (9). A five-point and three-time smoothing method was adopted in this study, and the number of smoothing iterations was 100. In the process of data smoothing, the signal usually contained the environment vibration information. To preserve the frequency band of the cable vibration, bandpass filtering was used to filter the signal. The initial parameters of the VMD algorithm were selected as $K = 3$, $\alpha = 2000$, and stop condition $\epsilon = 1 \times 10^{-6}$ The entire process was divided into the following steps. First, signal smoothing and fundamental frequency determination were conducted. Then, a bandpass filter for signal filtering was constructed based on the fundamental frequency. Then, the VMD algorithm was used to decompose the filtered signal and the Hilbert–Huang transform was applied to the 1st component to obtain the instantaneous frequency. Finally, the time-varying cable force was calculated according to the instantaneous frequency and the cable force-frequency formula as shown in Equation (16).

### 3.1.1 Recognition Results for Conditions 1 and 2

The operating conditions 1 and 2 were under constant tensile force of 5 kN and 6 kN, respectively. In each condition, the original signal of the cable vibration was obtained and smoothed to remove the interference signal during the acquisition process according to Equation (9). The ordinary and smoothed signal of condition 1 were shown in Figure 3 for an example. Then, a fast Fourier transform was performed on the smoothed signal to obtain the fundamental frequency by the frequency difference method, as shown in Figure 4. It was found that the fundamental frequency belonged to [4.33 Hz] and [4.05 Hz] for conditions 1 and 2 respectively.Therefore, a bandpass filter with a cut-off threshold of 4 Hz and a cut-off threshold of 6 Hz, namely, [4 Hz, 6 Hz], was constructed to filter the smoothed signal in two conditions. Then the VMD algorithm was performed on the filtered signal and the processed signals are shown in Figure 4. Obviously, the first component (IMF1) was fuller than the other two components (IMF2 and IMF3), and the value also showed a progressively decreasing form on the order of magnitude, indicating that IMF1 was a relatively major component. Taking the 1st component (IMF1) as the input, the Hilbert–Huang transform was used to obtain the instantaneous frequency curve. At last, the time-varying cable force value can be calculated from the instantaneous frequency value according to Equation (16). Comparing the calculated instantaneous cable force with the real cable force obtained by the tension sensor, the cable force of conditions 1 and 2 fluctuated around the real value of 5.0 kN and 6.0 kN respectively.

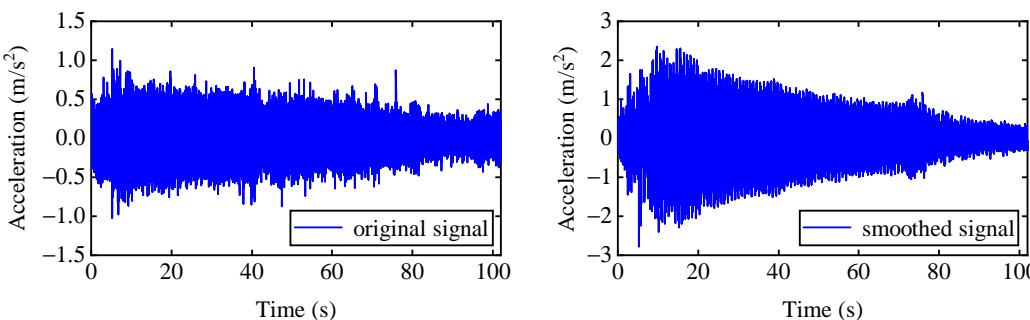

**Figure 3.** Condition 1 of scale model experiment: original and smoothed cable vibration signal processed by Equation (9).

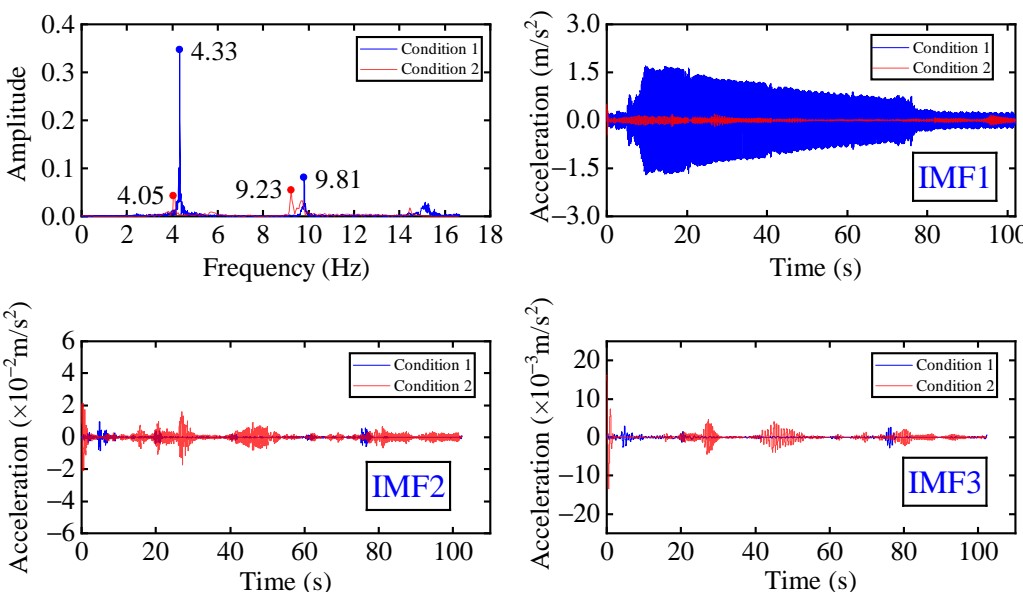

**Figure 4.** Conditions 1 and 2 of scale model experiment: the fast Fourier transform and variational mode decomposition (VMD) decomposition results.

### 3.1.2 Recognition Results for Conditions 3 and 4

Conditions 3 and 4 consider the situation where cable force may suddenly decrease and increase, respectively. For condition 3, 60 s vibration under a tensile force of F = 6 kN, a sudden reduction in the tensile force to 5.3 kN and 48 s vibration were conducted in turn, while for condition 4, the cable vibrated under the pulling force of F = 5 kN for 60 s and the pulling force was suddenly increased to F = 7.3 kN at the end of 60 s. The results of the Fourier change are shown in Figure 5, which can be seen that the fundamental frequency was in the [4.30 Hz] and [4.05 Hz] range for conditions 3 and 4 respectively. A bandpass filter with a cut-off threshold [4 Hz, 6 Hz] was constructed to filter the smoothed signals. The results of VMD decomposition are shown in Figure 5, and the time-varying cable force curves are shown in Figure 6. The proposed method can identify the start point and trend of sudden changes in cable force.

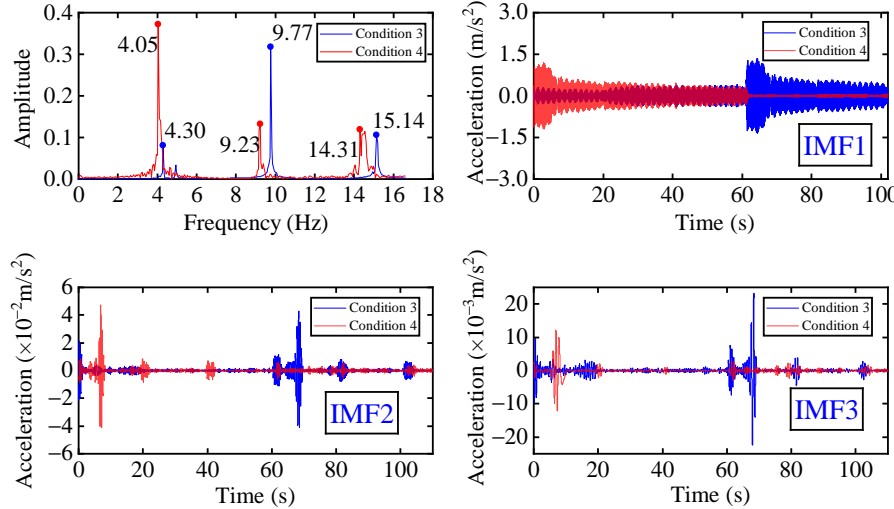

**Figure 5.** Conditions 3 and 4 of scale model experiment: the fast Fourier transform and VMD decomposition results.

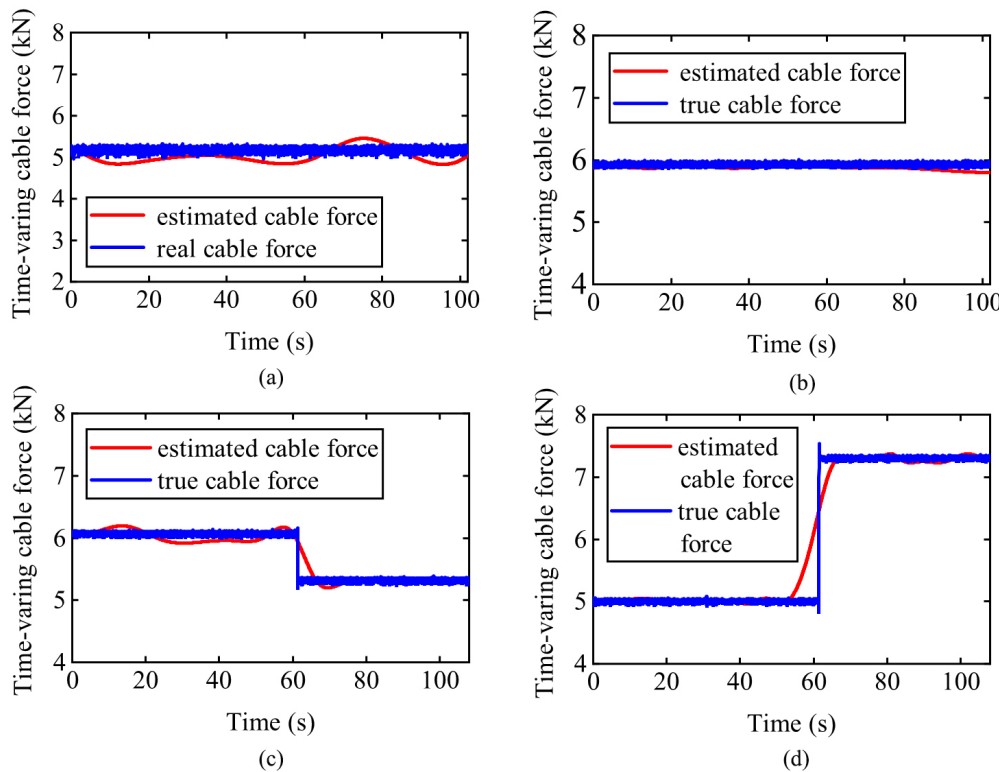

**Figure 6.** Time-varying  cable forces of scale model experiment: (**a**) Condition 1, (**b**) Condition 2, (**c**) Condition 3, (**d**) Condition 4.

### 3.2. Full-Scale Model Experiment

The feasibility of the proposed time-varying cable force identification method was preliminarily explained on the scale model. A full-scale test was also performed to further prove the feasibility of the proposed method in this paper. A single full-scale stay cable was produced to obtain the vibration acceleration data of the cable under different tensile conditions, and the VMD algorithm was used to identify the time-varying cable force. The full-scale model of the tested stayed cable is shown in Figure 7. Similar to the scaled experiment, the left part in this Figure the also the enlarged sketch of the tension and anchor. The length of this stayed cable was 167.85 m, the mass per unit length was 28.0 kg/m, the diameter was 87 mm, the elastic modulus EI = $1.97 \times 10^{11}$ Pa, the tensile strength of the cable was 1670 MPa, and the damping ratio and fundamental frequency were 0.012 and 0.84 Hz respectively.  As mentioned before, the proposed method only needs the acceleration information obtained by an acceleration sensor, and an additional sensor was also added for accurate data. Two acceleration sensors were installed at 38 m and 48 m, as shown in Figure 7. The sampling frequency of the sensors was 100 Hz. The whole layout is shown in Figure 8. In actual situations, sudden changes may occur in cable force, such as the damper falling. Therefore, three operating conditions were designed in this full-scale experiment to consider all situations, as shown in Table 2.

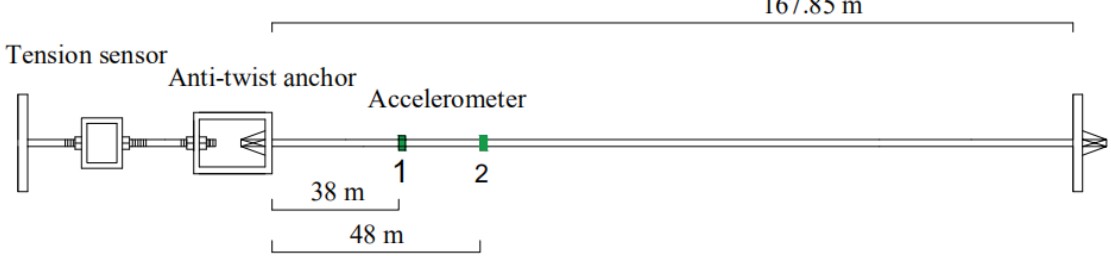

**Figure 7.** Design drawing of the stay cable in the full-scale model experiment.

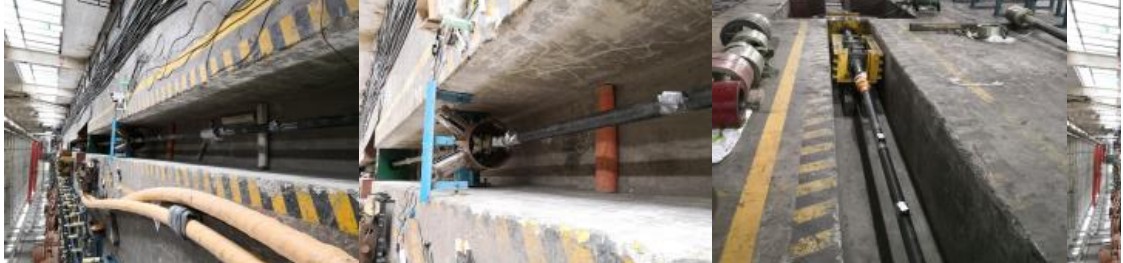

**Figure 8.** Field layout of the stay cable in the full-scale model experiment.

**Table 2.** Designed conditions information of the full-scale experiment.

| Conditions | Description |
|---|---|
| 1 | F = 2170 kN in t ∈ [0 s, 115 s] |
| 2 | F = 1635 kN in t ∈ [0 s, 115 s] F = 1903 kN in t ∈ [115 s, 230 s] |
| 3 | F = 2170 kN in t ∈ [0 s, 115 s] F = 1635 kN in t ∈ [115 s, 230 s] F = 1903 kN in t ∈ [230 s, 290 s] |

### 3.2.1. Recognition Results for Condition 1

For condition 1, the cable vibrated under the constant tensile force of 2170 kN for 115 s. The original signal and the signal after smoothing are shown in Figure 9. A fast Fourier analysis of the smoothed signal which shows the fundamental frequency information is shown in Figure 10. The frequency of the cable was identified by the peak value of the spectrum, and the fundamental frequency was calculated by the frequency difference method. Specifically, the average frequency of each calculated frequency difference was used to obtain the fundamental frequency of the cable. The fundamental frequency was calculated as 0.83 Hz. However, due to that the low-frequency components were not easy to capture, the multiplication frequency was considered. the low-frequency components were very dense, and it was not easy to select the first-order frequency. Therefore, the multiplication frequency was considered and the first-order frequency was determined by multiplying the average frequency of the intensive phase by the amplification factor, and then gradually select the frequency with prominent subsequent amplitude as the corresponding Frequency order. Since the third-order frequency was 2.49 Hz, the bandpass filtering with the filtering range [2.1 Hz, 2.8 Hz] was selected. The results of VMD decomposition are shown in Figure 11. After performing the Hilbert–Huang transform on IMF1, the time-varying cable forces are shown in Figure 12.

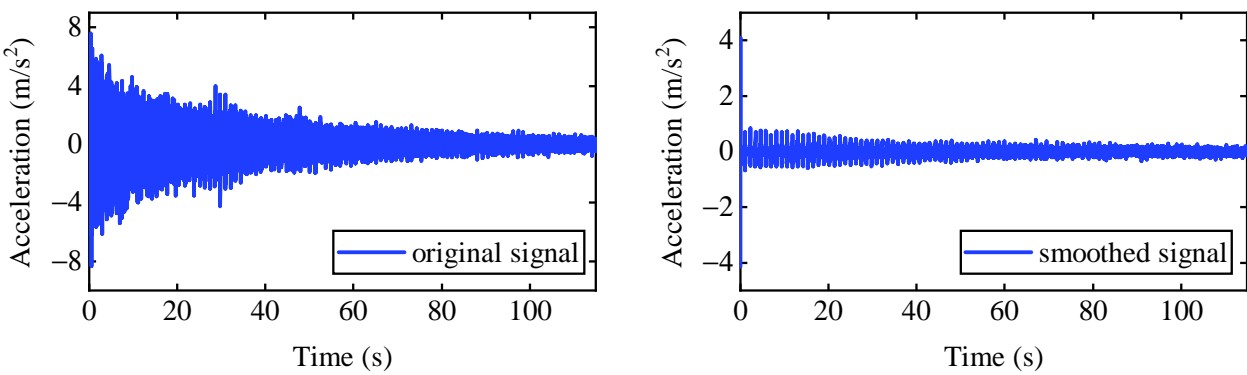

**Figure 9.** Condition 1 of the full-scale model experiment: original and smoothed cable vibration signal.

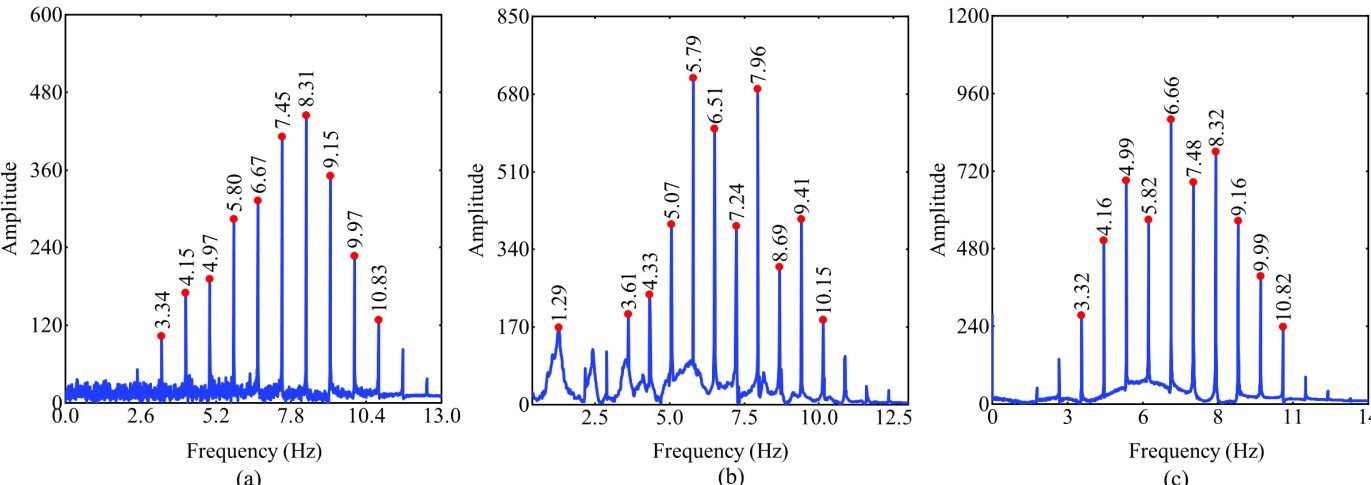

**Figure 10.** Fast Fourier transform results of the full-scale model experiment: (**a**) Condition 1, (**b**) Condition 2, (**c**) Condition 3.

### 3.2.2. Recognition Results for Condition 2

The cable first vibrated for 115 s under a constant tensile force of 1635 kN, then vibrated for 115 s under the tensile force of 1903 kN. A fast Fourier analysis of the smoothed signal shows that the fundamental frequency was approximately 0.73 Hz as shown in Figure 10, while the third-order frequency was 2.19 Hz and the bandpass filtering with the range [1.7 Hz, 2.5 Hz] was selected. The results of VMD decomposition are shown in Figure 11. The corresponding cable force are shown in Figure 12.

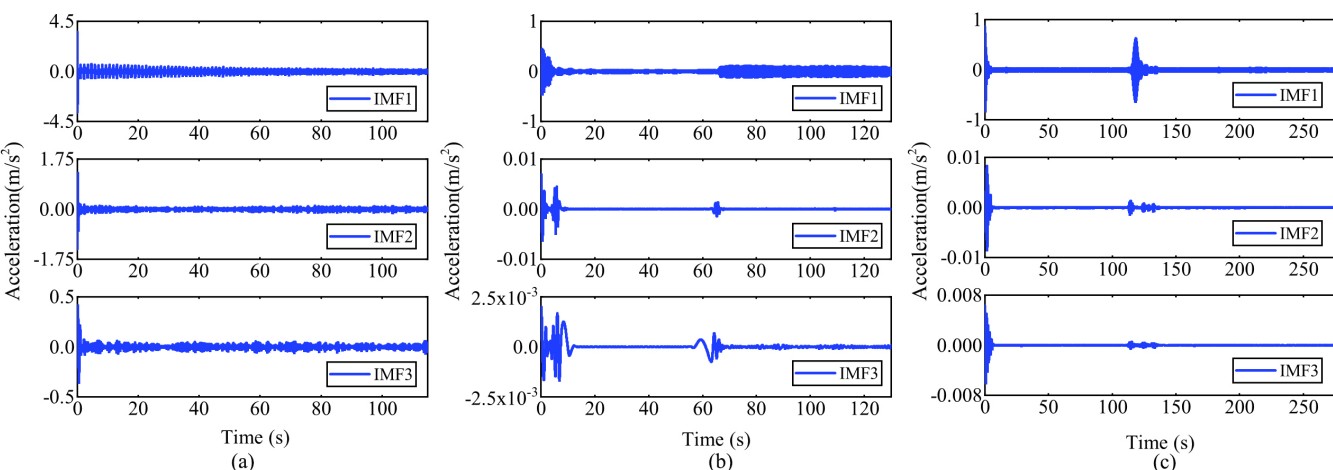

**Figure 11.** VMD decomposition results of the full-scale model experiment: (**a**) Condition 1, (**b**) Condition 2, (**c**) Condition 3.

### 3.2.3. Recognition Results for Condition 3

The cable vibrated for 115 s under the tensile force of F = 2170 kN, then vibrated for 115 s under F = 1635 kN, and vibrated for 60 s under F = 1903 kN at last. A fast Fourier analysis with fundamental frequency information is shown in Figure 10. The frequency was approximately 0.83 Hz, while the third-order frequency was approximately 2.50. The results of VMD decomposition are shown in Figure 11. The cable force are shown in Figure 12. It can be seen that the method proposed in this paper can identify the time moments of the sudden increase and decrease of cable force.

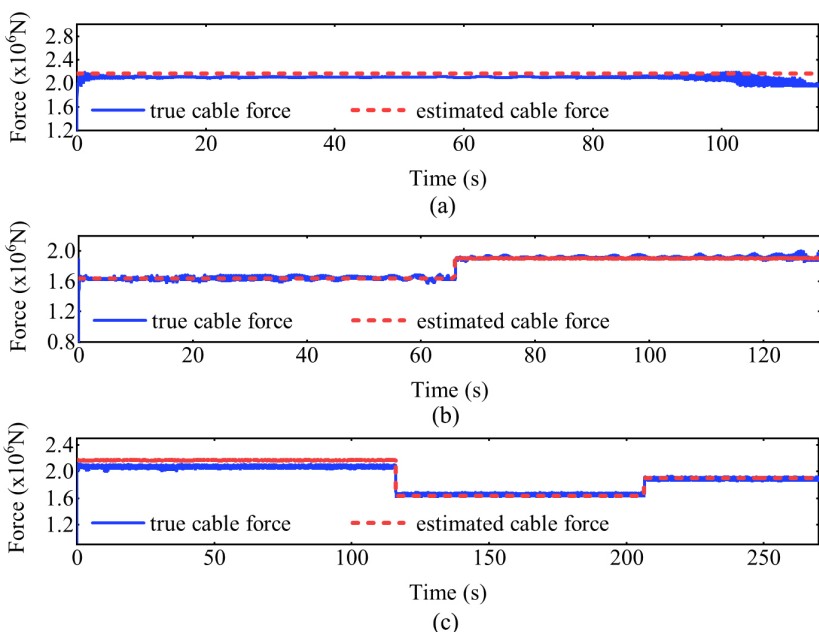

**Figure 12.** Time-varying cable forces of the full-scale model experiment: (**a**) Condition 1, (**b**) Condition 2, (**c**) Condition 3.

### 3.3. Error Analysis

From the above results, it can be seen that the proposed method identified the time point and trend in the sudden increase and decrease in the cable force, which coincides with the true value of the cable force in the interval stability stage. It can been seen from Figures 6 and 12 that there was a time delay of estimated force for condition 3 and 4 in scaled model experiment, and there was little delay in full-scale model experiment. This may be caused by the mass of sensor, the force of steel wire in scaled model can be influenced by the accelerator while there may be little influence considering the large size and weight of full-scale model experiment which selected the real cable. Although the sensitivity of force change is different, the cable force value and sudden change identified by this two experiments were consistent with the experimental design. Errors were calculated according to the stationary stage and the mutation, and the relative error was used for calculation. The results are shown in Tables 3 and 4. It can be seen from the table that the proposed method had a better recognition effect in the middle part of the signal. The errors of the scale model and the full-scale model in the stationary stage were both below 6%, and the average error was also below 5%. In the scale experiment, the error of mutation points was relatively large. The main reason is that when a sudden force is applied, it is adjusted with a manual wrench, which interferes with the vibration of the cable, but all of them identified the mutation trend very well. The error at the mutation points is relatively small because the abrupt force applied by the hydraulic device is actually smoother and automatic.

**Table 3.** Scale experiment error statistical table.

| Conditions | Maximum Stationary Phase Error (%) | Maximum Mutation Point Error (%) | Average Error (%) |
|---|---|---|---|
| 1 | 8.941 | / | 4.243 |
| 2 | 3.189 | / | 0.825 |
| 3 | 3.542 | 13.939 | 2.022 |
| 4 | 2.563 | 29.465 | 1.504 |
| mean | 4.753 | 21.702 | 2.067 |

Table 4. Full-scale experimental error statistics table.

| Conditions | Maximum Stationary Phase Error (%) | Maximum Mutation Point Error (%) | Average Error (%) |
|---|---|---|---|
| 1 | 8.009 | / | 3.479 |
| 2 | 2.942 | 4.066 | 0.971 |
| 3 | 4.864 | 7.369 | 2.288 |
| mean | 5.272 | 5.718 | 2.246 |

## 4. On-Site Bridge Experiment

### 4.1. Experimental Introduction and Layout

The Poyang Lake Bridge in Jiangxi, China, was selected as the experimental bridge, which has a total length of 3799 m and a deck width of 27.5 m. In addition, the length of the main bridge is 636 m, and the main bridge consists of four spans that are arranged across the border as 65 m + 123 m + 318 m + 130 m, as shown in Figure 13. In July 2019, during a routine inspection of the bridge, it was found that four bolts on the outside of the tensile and compression support on the downstream side of the auxiliary pier were broken. One week later, 4 bolts on the inner side of the tensile and compression support on the downstream side of the auxiliary pier broke and fell off, as shown in Figure 14. The failure of the steel plate anchor bolts on the downstream compression bearing caused the bearing to vacate and quit work. After a period of monitoring, the downstream support of the auxiliary pier was replaced from 10 October 2019, to 12 October 2019. During this period, auxiliary cable force monitoring was carried out on the cable numbered A15 on the downstream support.

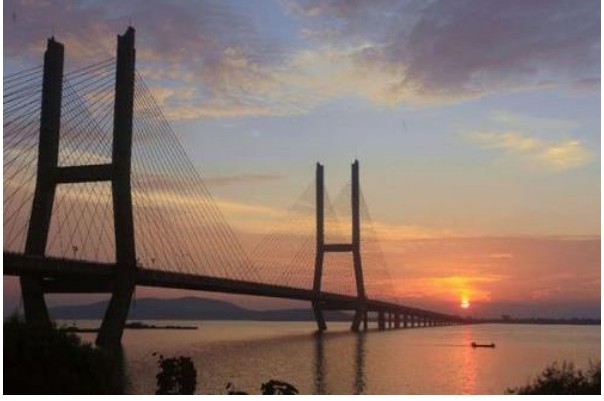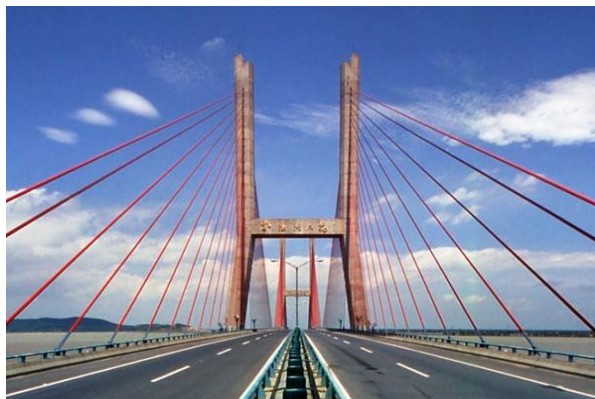

Figure 13. the overview of Poyang bridge.

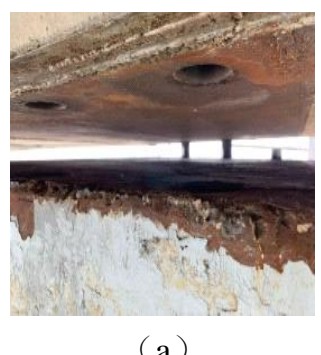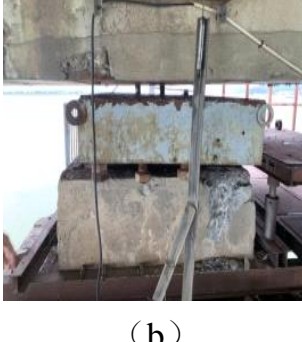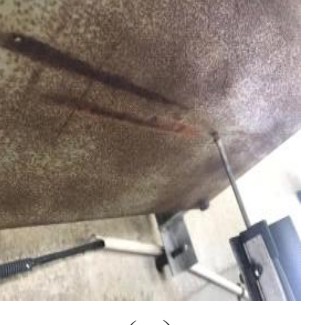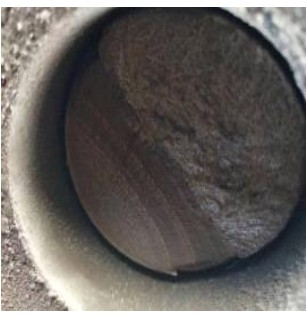

（a） （b） （c） （d）

Figure 14. Schematic diagram of support failure: (**a**) the bolt is off, (**b**) the cushion stone is damaged, (**c**) the lateral displacement (**d**) the broken bolt.

The length of the A15 cable is 146.92 m, the weight per unit length is 60.1 (kg/m), the diameter is 125 mm, and the cable force of completion is 430 (t). When arranging on site, the acceleration sensor was placed at a distance of approximately 3 m from the bottom of the cable, and the sampling frequency was set to 500 Hz. Three working conditions were selected for this experiment; specifically, the state after the damaged support on the downstream side was removed, the state where the new support was installed, and the bolts were tightened, and the state after the new support was installed. Due to the high sampling frequency of the sensor and the large quantity of data, it required approximately 400 s to calculate the cable force per hour for the first two conditions, as shown in Table 5. The condition 2 indicates the repairing process, and the whole bridge stress was changing all the time. The condition 3 indicates that the whole repairing process finished, and the new stress state of whole bridge including the cable forces was stable.

**Table 5.** Working conditions and the corresponding time period.

| Condition | Description | Corresponding Time Period |
|---|---|---|
| 1 | After removing the damaged support on the downstream side | 11 October 2019 10:10–10:20 |
| 2 | Install new support and tighten bolts | 11 October 2019 15:00–16:00 |
| 3 | After the new support is installed | 12 October 2019 09:00–09:30 |

*4.2. Experimental Results and Analysis*

4.2.1. Recognition Results for Condition 1

Condition 1 was in a state after the broken support was completely removed. The time interval of 10:10–10:20 of the acceleration signals were selected for calculation on 11 October 2019. The acceleration signal and the frequency spectrum are shown in Figure 15, a period of 400 s of acceleration data was selected for calculation. The fundamental frequency basically changed in the range of 0.76 Hz to 0.79 Hz. The cable force results calculated based on the variational mode decomposition and the HHT method are shown in Figure 15. The cable force fluctuated relatively in the range of [$3.5 \times 10^3$ kN, $4 \times 10^3$ kN]. The main reason is that the side beam was unconstrained after the downstream support was removed.

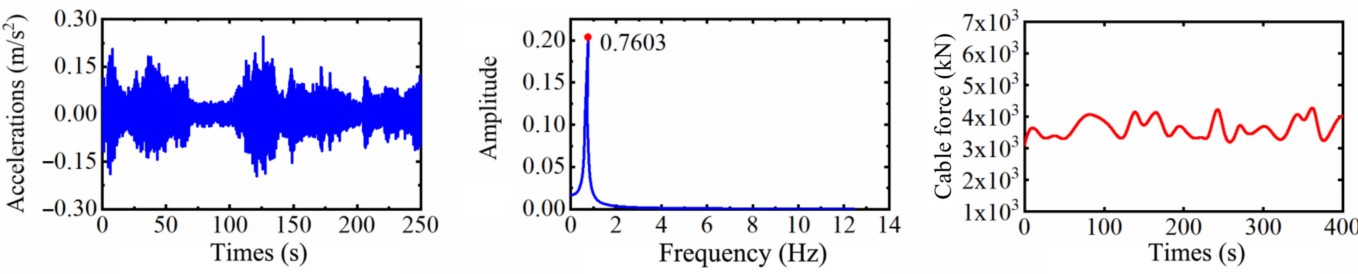

**Figure 15.** Acceleration signal, vibration spectrum and time-varying force of the A15 cable in the time interval of 10:10–10:20 on 11 October 2019.

4.2.2. Recognition Results for Condition 2

Condition 2 was in a state where a new support was installed, and the bolts were gradually tightened. In the time zone of 15:00–16:00 on 11 October 2019, a period of 400 s of the acceleration signals were selected for calculation. The acceleration signal and the frequency spectrum are shown in Figure 16. The fundamental frequency basically changed in the range of 0.78 Hz to 0.88 Hz. The cable force results based on the variational mode decomposition and HHT method are shown in Figure 16. The cable force gradually increased from $3.8 \times 10^3$ kN to $4.3 \times 10^3$ kN, but the fluctuation was better than working condition 1 because the beam was gradually restricted, and the growth of the three time periods from 14:00–17:00 was relatively obvious, from $3.8 \times 10^3$ kN to $4.0 \times 10^3$ kN. Time

period 17:00–18:00 was about to be completed, and the force basically fluctuates within the range of $[4.0 \times 10^3 \text{ kN}, 4.3 \times 10^3 \text{ kN}]$ except for local points.

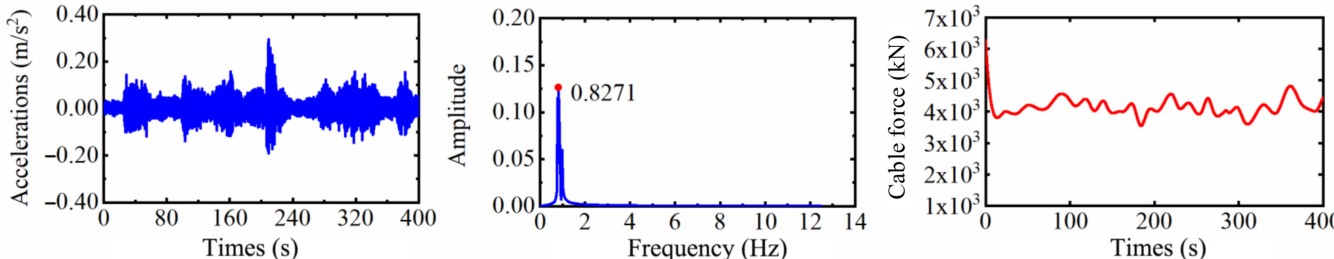

**Figure 16.** Acceleration signal, vibration spectrum and time-varying force of the A15 cable in the time interval of 16:00–17:00 on 11 October 2019.

### 4.2.3. Recognition Results for Condition 3

Condition 3 was in a state after the installation of the new support was completed, and it was the next morning, so it was basically stable. The acceleration data with a duration of approximately 160 s were selected for analysis. The acceleration signal diagram and the fundamental frequency are shown in Figure 17. According to the inspection report of past years, the completed cable force of A15 was 4300 kN, the cable force measured in December 2015 was 4480 kN, the cable force measured in April 2017 was 4450 kN, and the cable force measured in March 2018 was 4500 kN. As shown in Figure 17, though the proposed method has achieved good results in the lab experiments, it is found that there was a certain error in the on-site bridge measurement. The identified cable force value fluctuates around 4475.0 kN which was close to the values measured in previous years.

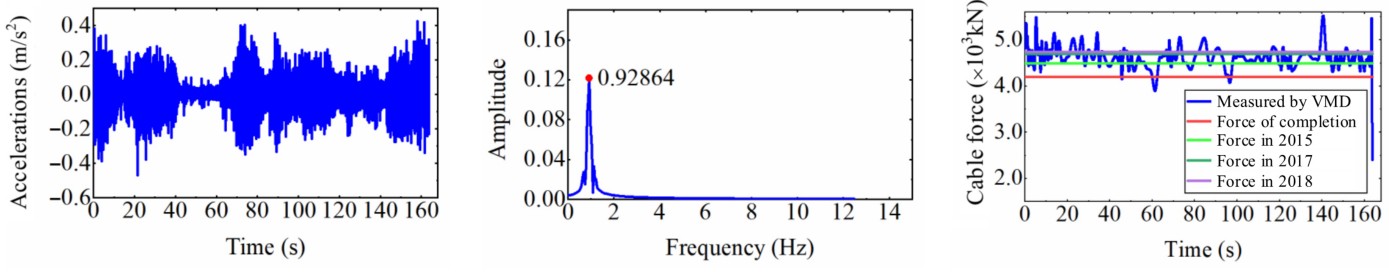

**Figure 17.** Acceleration signal, vibration spectrum and time-varying force of the A15 cable in the time interval of 09:00–09:30 on 12 October 2019.

It can be seen from the above results that the time-varying cable force identification method based on variational modal decomposition and HHT can basically identify the cable force change trend and can be used as an auxiliary monitoring method during the replacement of the support.

### 5. Conclusions

In this article, a new mothod based on VMD was established to identify the time-varying force of stay cables. VMD is a new signal decomposition method, which searches for the optimal solution of the constrained variational model established by the specified center frequency to achieve adaptive signal decomposition. Due to its advantages, VMD has been used by researchers in the field of damage diagnostics and signal decomposition. The advantage of the Hilbert–Huang transform (HHT) is that it can find the instantaneous frequency of the signal and can give a physical meaning to the instantaneous frequency of the signal. This paper analyzed the theory and implementation steps of the VMD algorithm, and HHT transformation was also integrated to show that both methods can be used to calculate the instantaneous frequency of each order mode. Then, a scaled

stay cable model, a full-scale stay cable model and an on-site bridge experiment were conducted to validate the proposed method. The VMD and HHT methods were applied to the vibration acceleration signal of the stay cables, the instantaneous frequency of the first-order vibration mode of the stay cable was obtained, and the instantaneous cable force based on the subinstant frequency was calculated and compared with the real cable force. The results show that the method in this paper can successfully identify the time-varying cable force even when the mutation amplitude is more than 15%, when using one accelerometer without extra information. For the initial vibration interval, the signal contains less information so that the initial value of the iteration has greater impact, which may cause fluctuations.

The errors of the scaled model and the full-scale model in the stationary stage are both below 6%, and the average error is also below 5%; there are fluctuations at the mutation point, the average error of the scaled experiment mutation is 21.702%, and the average error of the full-scale experiment mutation is 5.718%, but all of them can identify the mutation point very well, indicating that the time-varying cable force identification method based on the variational mode decomposition method has a better identification effect for stable signals, and the error at the mutation intervals is relatively large but can be identified as discontinuity. In general, the method in this paper can identify the time-varying cable force well in the experiments and the on-site test, the algorithm in this paper can be used to identify time-varying cable forces in actual engineering. Further research should be conducted focus on more on-site bridge tests for the validation and improvement of the time-varying cable force identification methods.

**Author Contributions:** Conceptualization, G.W.; Methodology, S.H.; Validation and formal analysis, B.D.; Investigation, J.F. and Y.H.; Data resources, H.W. and X.Z. All authors have read and agreed to the published version of the manuscript.

**Funding:** This research is funded by the National Natural Science Foundation of China (Grant No. 51525801), Graduate Research and Innovation Projects of Jiangsu Province, China (No. KYCX17_0120) and the Shanxi Provincial Transportation Science and Technology Project (Grant No. 2019-1-19; 2019-2-4.

**Institutional Review Board Statement:** This research is studying the cable force identification of bridges, the IRB review of this research would not be required.

**Informed Consent Statement:** The permission of all materials including sensor data, images, videos, etc are granted to be used by researchers for scientific publications.

**Data Availability Statement:** Not applicable.

**Acknowledgments:** The authors greatly appreciate the financial support from the National Natural Science Foundation of China (Grant No. 51525801), Graduate Research and Innovation Projects of Jiangsu Province, China (No. KYCX17_0120) and the Shanxi Provincial Transportation Science and Technology Project (Grant No. 2019-1-19; 2019-2-4).

**Conflicts of Interest:** We declare that we do not have any commercial or associative interest that represents a conflict of interest in connection with the work submitted.

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
