# Peer review of "Variational Mode Decomposition Based Time-Varying Force Identification of Stay Cables"

_applsci, doi:10.3390/app11031254_

Round 1

Reviewer 1 Report

Dear Authors,

thank you for this interesting paper. Please find below my comments that can be used to improve your manuscript.

ABSTRACT

  1. This chapter is quite clear, it describes what has been done and what has been found.

INTRODUCTION

  1. This chapter is quite clear, it describes the state of the art and positions the proposed method within the research area.

VARIATIONAL MODE DECOMPOSITION ANALYSIS

  1. This chapter is quite clear, it describes the method used in the research.

STAY CABLE EXPERIMENTS

  1. Can you please elaborate on "The distance between the anchoring ends of the steel wire was 15 m..." (lines 85 and 86)? What about the rest of the system - from these 15 m to the anchor plate. Indicating that 15 m is the cable length is quite questionable.
  2. Can you describe the boundary conditions that you have achieved in your experiment (hinge, clamp...)?
  3. Can you provide some basic equipment specification (load cell, accelerometer...). The accelerometer seems rather massive and that influences the measurement results. What was the anti-twist anchor, it is not present on figures provided? 
  4. How did the blower excite the cable? From Figure 2, it seems that the excitation was in horizontal plane, and you measured the acceleration in vertical plane. You have also mentioned a baffle that you used, but it is not visible from the figure. Can you please elaborate on this?
  5. According to the above comments, please consider modifying Figures 1 and 2.
  6. Line 97 - please add "change" or a similar word here. Please describe how did you induce this sudden force change. Namely, Table 1 (time intervals) suggests that you have induced it in less then 1 s.
  7. Figure 3 - please explain what caused the acceleration increase in the smoothed signal.
  8. Figure 4 - please explain why are the frequencies for condition 2 lower then for condition 1 when the force is higher in case of condition 2. 
  9. Can you please provide Hilbert-Huang transform record (instantaneous frequency curve)?
  10. Figure 5 - for conditions 3 and 4 you have identified 4,30 Hz and 4,05 Hz frequencies. This was before the sudden force change of afterwards? Please elaborate on this.
  11. Line 148 - EI is flexural/bending stiffness/rigidity. Please add measurement unit.
  12. Line 149 - fundamental frequency must have a measurement unit. What do you consider under "fundamental frequency", the frequency under what conditions?
  13. Regarding the full-scale model experiment setup, please address my previous comments regarding the scale model experiment (i. e. comments 1 to 6). 
  14. Lines 163 to 167 - why were the low-frequency components hard to capture? Please specify the measurement equipment data (address comment 3). What do you consider under "multiplication frequency"? How did you determine the "third-order frequency", how did you define it?
  15. Figure 10 - please see comment 10. Please elaborate on this comment from the aspect of conditions 2 and 3.
  16. Line 172 - how did you determine the 2,19 Hz frequency?
  17. Line 179 - how did you determine the 2,50 Hz frequency?
  18. Table 3 is presenting 5 conditions and you had 4 conditions in the experiment.

REAL BRIDGE EXPERIMENT

  1. Figure 14 is pretty unclear. It is quite small and the camera angle is makes it hard to visualize the whole system. Can you improve this? The same comment is valid for Figure 8 from the previous chapter.
  2. Lines 210 to 212 - can you describe the difference between conditions 2 and 3? Namely, after the new support is installed and the bolts are tighten, the reparation procedure seems to be finished. So, what does the next condition "after the new support is installed" mean?
  3. What equipment did you use for this test, can you provide some specifications?
  4. Did you excite the cable? If yes, please describe how.
  5. Line 220 - this range does not comply with the 0,7603 Hz frequency indicated in Figure 15.
  6. Figure 15 - condition 1 lasted for 600 s. Please position this 250 s record in that range since currently it is not clear whether this record belong to the beginning of the condition, end of the condition or somewhere in the middle. The spectrum record - what time block did you analyze to get this spectrum? The cable force record - the same comment as for the acceleration record can be applied (please position this record in the 600 s long condition). Please not that the time blocks for acceleration and cable force do not have equal length. Why?
  7. Figure 16 - same comments as for Figure 15. The overall time block for condition 2 is 36000 s and you need to position these records within that block.
  8. Line 232 - you are indicating time periods from 14:00-17:00 and 17:00-18:00. Please explain this since you indicated before that condition 2 lasted from 15:00-16:00.
  9. Figure 17 - same comments as for Figures 15 and 16. The overall time block for condition 3 is 18000 s and you need to position these records within that block.
  10. Lines 240 and 241 - how were the forces measured during the indicated inspections? Also, when was the first measurement (430 kN) conducted - you did not indicate when was the bridge built. Also, please note that your previous text and cable force records indicate values that are 10 x bigger than the values indicated in these lines.
  11. Please note that according to Figure 17, the cable force changes are significant (more then 20 %). Please elaborate on this since this presents a very high measurement uncertainty. Your estimated value of 447,50 kN - how did you get it?

CONCLUSION

  1. Line 259 - was it first-order vibration mode in all cases? In the text, you have also mentioned third-order modes.
  2. Line 263 - "even when the mutation amplitude is more than 15%". What do you mean by this? This change is quite big. Indicating that you can detect such a big change (and even a bigger one) seems inappropriate.
  3. Line 273 - please explain, "better" in comparison with some other methods or what?
  4. Lines 273 to 276 are basically repeating some of the previous sentences in this chapter.
  5. Why didn't you mention real bridge experiment in your conclusions?
  6. Please try to organize your conclusions in a more concise manner. Currently, I don't see a reason why should one use this method instead of the classical FFT where the obtained frequencies can be used for force calculation (using the eqs. that you indicated in your paper).

Author Response

Response to Reviewer 1 Comments

Dear Editors and Reviewers:

Thank you for the valuable and helpful comments for revising and improving the quality of our paper. We have studied the comments carefully and have made the corresponding correction. Revised portion were marked in blue in the paper. The responses to the reviewer’s comments are as following:

ABSTRACT

Point 1: This chapter is quite clear, it describes what has been done and what has been found.

Response 1:

The authors are very grateful for your comment.

INTRODUCTION

Point 1: This chapter is quite clear, it describes the state of the art and positions the proposed method within the research area.

Response 1:

The authors appreciate the reviewer’s comments very much.

VARIATIONAL MODE DECOMPOSITION ANALYSIS

Point 1: This chapter is quite clear, it describes the method used in the research.

Response 1:

Thank you very much for your review.

STAY CABLE EXPERIMENTS

Point 1: Can you please elaborate on "The distance between the anchoring ends of the steel wire was 15 m..." (lines 85 and 86)? What about the rest of the system - from these 15 m to the anchor plate. Indicating that 15 m is the cable length is quite questionable.

Response 1:

The author is very grateful for your very good question.

As Figure 1 shows, the rest of the system was the tension sensor and the anti-twist anchor, while two accelerometers were fixed on the cable. The left part of Figure 1 is an enlarged picture of the details, and the actual situation was not so long.

Point 2: Can you describe the boundary conditions that you have achieved in your experiment (hinge, clamp...)?

Response 2:

The boundary condition is that one end is fixed and the other is simply-supported.

Point 3: Can you provide some basic equipment specification (load cell, accelerometer...). The accelerometer seems rather massive and that influences the measurement results. What was the anti-twist anchor, it is not present on figures provided?

Response 3:

The author would like to thank you for your suggestions

The anti-twist anchor aims to restrain the twist of the cable, which did not appear in the figures due to the shooting angles of the figures.

The specification of the basic equipment is as follows:

  • The type of acceleraometer is 941B which is a commercial sensor used in many lab and on-site experiments. The weight is 750g. As the commercial acceleration sensors have the standard weight (like 500-800g), the influence on the results can not be removed. As the methods in this paper mainly intend to validate the VMD and HHT in the time-varying cable force identification and sudden change, the scaled-model experiment results in this paper achieved this target. The weight influence on the lab experiment of this field will be deeply studied in the further research.
  • The load was applied by the jack alongside the cable wire, and the tensor load sensor was chosen the commercial sensor with the range of [0,10KN]

Point 4: How did the blower excite the cable? From Figure 2, it seems that the excitation was in horizontal plane, and you measured the acceleration in vertical plane. You have also mentioned a baffle that you used, but it is not visible from the figure. Can you please elaborate on this?

Response 4:

Thanks for these suggestions.

The blower caused the cable to vibrate by blowing winds, so that the cable was excited. The accelerometer we used can measure acceleration in the horizontal direction. The baffle was used to block the winds and help the cable to vibrate with higher amplitude.

Point 5: According to the above comments, please consider modifying Figures 1 and 2.

Response 5:

The tension sensor and Anti-twist anchor in Figure 1 has been mentioned in the revised paper seen in L.87.

Figure 2 has been modified according to the comments.

Point 6: Line 97 - please add "change" or a similar word here. Please describe how did you induce this sudden force change. Namely, Table 1 (time intervals) suggests that you have induced it in less then 1 s.

Response 6:

The word “change” has been added in the paper. The sudden force change was induced by the cable force adjustment device in the Figure 2, the sudden force can be changed with the jack, and the specific force value can be acquired by the tension sensor.

Point 7: Figure 3 - please explain what caused the acceleration increase in the smoothed signal.

Response 7:  

The author is very grateful for your very good question. As mentioned in L.101-L.103, and the collected original signals may generate a large number of “burrs”. A five-point and three-time smoothing method was adopted in this study, the figure 3 shows results of the “burrs” signal and smoothed signals, these two signals are chosen to show the comparison results, but two different signal time zones were chosen.

Point 8: Figure 4 - please explain why are the frequencies for condition 2 lower then for condition 1 when the force is higher in case of condition 2.

Response 8:

The fundamental frequency should be different in these two conditions. As the fundamental frequency was selected by the first peak, the corresponding frequency in condition 2 is lower than the that in condition 1. These may be caused by the stiffness change of the whole devices including cable, sensor and anchors. This phenomenon has also been seen in the bridge beam damage detection, the damaged beam has higher stiffness values than the undamaged beam.

Point 9Can you please provide Hilbert-Huang transform record (instantaneous frequency curve)?

Response 9:

The author is very grateful for your very good question.

The Hilbert-Huang transform record was not provided in the paper due to the length of this paper. The figures are shown in the below.

Point 10Figure 5 - for conditions 3 and 4 you have identified 4,30 Hz and 4,05 Hz frequencies. This was before the sudden force change of afterwards? Please elaborate on this.

Response 10:

Sorry for the unclear explanation. The 4.30 Hz and 4.05 Hz were the frequencies before the sudden force change. These two results are the same with condition 1&2.  The Figure 4&5 show the IMF1 was fuller than the other two components (IMF2 and IMF3), and the value also showed a progressively decreasing form on the order of magnitude, indicating that IMF1 was a relatively major component. The following time-varying force will be obtained based on the IMF1 values.

Point 11Line 148 - EI is flexural/bending stiffness/rigidity. Please add measurement unit.

Response 11:

The authors appreciate the reviewer’s careful comments very much.

The unit was Pa. It has been modified in the paper.

Point 12: Line 149 - fundamental frequency must have a measurement unit. What do you consider under "fundamental frequency", the frequency under what conditions?

Response 12:

The authors appreciate the reviewer’s careful comments very much.

The unit of the "fundamental frequency" was Hz, which has been added in the paper. The fundamental frequency means the natural frequency.

Point 13: Regarding the full-scale model experiment setup, please address my previous comments regarding the scale model experiment (i. e. comments 1 to 6).

Response 13:

The author is very grateful for your very good question.

The comments regarding the full-scale model experiment has been processed as the same as that of the scaled model experiment. The cable was an original actual cable that will be used on the bridges.

Point 14: Lines 163 to 167 - why were the low-frequency components hard to capture? Please specify the measurement equipment data (address comment 3). What do you consider under "multiplication frequency"? How did you determine the "third-order frequency", how did you define it?

Response 14:

As can be seen in Figure 10(a), the low-frequency components were very dense, and it was not easy to select the first-order frequency. Therefore, the first-order frequency was determined by multiplying the average frequency of the intensive phase by the amplification factor, and then gradually select the frequency with prominent subsequent amplitude as the corresponding Frequency order.

Point 15: Figure 10 - please see comment 10. Please elaborate on this comment from the aspect of conditions 2 and 3.

Response 15:

Sorry for the unclear explanation again. The frequencies in Figure 10 (b) and (c) were calculated before the sudden force change. These two results show the frequency distribution in different force conditions which applied on the full scale cables. As can be seen in Figure 11, the IMF1 was fuller than the other two components (IMF2 and IMF3), and the value also showed a progressively decreasing form on the order of magnitude, indicating that IMF1 was a relatively major component. The following time-varying force will be obtained based on the IMF1 values.

Point 16: Line 172 - how did you determine the 2,19 Hz frequency?

Response 16:

The authors appreciate the reviewer’s careful comments very much.

It was our negligence that we did not give the figure due to the length of the paper. The 2,19Hz frequency was obtained from the figure below.

Point 17: Line 179 - how did you determine the 2,50 Hz frequency?

Response 17:

The authors appreciate the reviewer’s careful comments very much.

It was our negligence that we did not give the figure due to the length of the paper. The 2,50Hz frequency was obtained from the figure below.

Point 18: Table 3 is presenting 5 conditions and you had 4 conditions in the experiment.

Response 18:

The authors appreciate the reviewer’s careful comments very much.

It was our negligence that there was some mistake in the Table 3. It has been modified.

REAL BRIDGE EXPERIMENT

Point 1: Figure 14 is pretty unclear. It is quite small and the camera angle is makes it hard to visualize the whole system. Can you improve this? The same comment is valid for Figure 8 from the previous chapter.

Response 1:

The author would like to thank you for your suggestions.

Sorry for the unclear images. As the real bridge experiment was conducted during the bearing repairing process. The damage scenarios near the bearings were presented. Since the experiment was done last year, these images were took when the authors climbed to the bottom of beam of this super bridge at that time. Sorry for not being able to provide a clearer picture.

Point 2: Lines 210 to 212 - can you describe the difference between conditions 2 and 3? Namely, after the new support is installed and the bolts are tighten, the reparation procedure seems to be finished. So, what does the next condition "after the new support is installed" mean?

Response 2:

The authors appreciate the reviewer’s careful comments very much.

The condition 2 means the repairing work was still conducted, the stress state of the whole bridge was changing all the time during this repairing process. The condition 3 means the whole repairing process was finished, and the stress state of the whole bridge including the cable force was stable.

Point 3: What equipment did you use for this test, can you provide some specifications?

Response 3:

The authors appreciate the reviewer’s careful comments very much.

The authors only used the acceleration sensors and corresponding data collecting device for this test as shown in the figure, the equipment used for data collection is a 24-bit network intelligent collector.

Point 4: Did you excite the cable? If yes, please describe how.

Response 4:

The author would like to thank you for your suggestions.

The authors did not excite the cable. The cable was excited by the ambient vibration. The cars on the bridge also excite the cable.

Point 5: Line 220 - this range does not comply with the 0,7603 Hz frequency indicated in Figure 15.

Response 5:

The authors appreciate the reviewer’s careful comments very much.

It was our negligence. The correct frequency should be [0.76Hz, 0.79Hz], which has been modified in the paper. You can see the figure below for details.

Vibration spectrum of the A15 cable in the time interval of 09: 30-10: 40 on October 11, 2019.

Point 6: Figure 15 - condition 1 lasted for 600 s. Please position this 250 s record in that range since currently it is not clear whether this record belong to the beginning of the condition, end of the condition or somewhere in the middle. The spectrum record - what time block did you analyze to get this spectrum? The cable force record - the same comment as for the acceleration record can be applied (please position this record in the 600 s long condition). Please not that the time blocks for acceleration and cable force do not have equal length. Why?

Response 6:

In this condition, we only selected a period of 400 seconds of acceleration data for calculation and show in this manuscript. There is a label mistake in the first figure in Figure 15. The time period has been modified.

Point 7: Figure 16 - same comments as for Figure 15. The overall time block for condition 2 is 36000 s and you need to position these records within that block.

Response 7:

The author would like to thank you for your suggestions.

We only selected a period of 400 seconds of acceleration data for calculation, not the entire time period of data used for calculation.

Point 8: Line 232 - you are indicating time periods from 14:00-17:00 and 17:00-18:00. Please explain this since you indicated before that condition 2 lasted from 15:00-16:00.

Response 8:

The authors collected all the data during 14:00-18:00. As can be seen in the Figure below.

Point 9: Figure 17 - same comments as for Figures 15 and 16. The overall time block for condition 3 is 18000 s and you need to position these records within that block.

Response 9:

The authors appreciate the reviewer’s careful comments very much.

It was our negligence. The related content has been modified. In fact, this condition means the repairing process finished, and the whole bridge and cable force was stable. The time period chosen in this manuscript can indicated the cable force change.

Point 10: Lines 240 and 241 - how were the forces measured during the indicated inspections? Also, when was the first measurement (430 kN) conducted - you did not indicate when was the bridge built. Also, please note that your previous text and cable force records indicate values that are 10 x bigger than the values indicated in these lines.

Response 10:

The authors appreciate the reviewer’s careful comments very much.

The cable force was monitored by the local management department. The bridge was built in 2000. The cable force should be 4300kN, 4480 kN, 4475kN. It was our negligence.

Point 11: Please note that according to Figure 17, the cable force changes are significant (more then 20 %). Please elaborate on this since this presents a very high measurement uncertainty. Your estimated value of 447,50 kN - how did you get it?

Response 11:

The authors appreciate the reviewer’s careful comments very much.

The method we proposed has achieved good results in the scaled and full-scale experiments, but it is found that there is a certain error in the real bridge with the cable force obtained by the actual measurement. We are currently studying the cause of this error. 447.5kN comes from our actual measurement data.

CONCLUSION

Point 1: Line 259 - was it first-order vibration mode in all cases? In the text, you have also mentioned third-order modes.

Response 1:

The authors appreciate the reviewer’s careful comments very much.

It is not necessarily the first-order vibration mode in all cases. In this paper the first three orders of the vibration were obtained by VMD, and the main order(the first-order in most cases) was chosen to calculate the cable force by HHT.

Point 2: Line 263 - "even when the mutation amplitude is more than 15%". What do you mean by this? This change is quite big. Indicating that you can detect such a big change (and even a bigger one) seems inappropriate.

Response 2:

The author would like to thank you for your suggestions.

In the scale and full-scale experiments, we have set the working conditions where the cable force changes suddenly (the mutation amplitude is more than 15%), and we want to verify the accuracy of the method that we proposed under this working condition.

Point 3: Line 273 - please explain, "better" in comparison with some other methods or what?

Response 3:

The authors appreciate the reviewer’s careful comments very much.

"better" was in comparison with the conventional methods that calculate the cable force. The vibration-based method, which calculates cable force by the cable force-fundamental frequency formula, the corresponding results can only provide the average force while the time-varying force result can be provided in this manuscript.

Point 4: Lines 273 to 276 are basically repeating some of the previous sentences in this chapter.

Response 4:

The author would like to thank you for your suggestions.

This part was discussing the feasibility of the method from the aspect of actual application.

Point 5: Why didn't you mention real bridge experiment in your conclusions?

Response 5:

The authors appreciate the reviewer’s careful comments very much.

It was our negligence that we did not mention that the real bridge experiment. We have added the related content.

Point 6: Please try to organize your conclusions in a more concise manner. Currently, I don't see a reason why should one use this method instead of the classical FFT where the obtained frequencies can be used for force calculation (using the eqs. that you indicated in your paper).

Response 6:

The author would like to thank you for your suggestions.

The innovation of our proposed method lies in the combination of VMD and HHT, which can provide instance cable force and find the accurate cable force changes over the experiment and on-site monitoring process.

Reviewer 2 Report

This article addresses an important topic in vibration and health monitoring analysis of cable stay bridges, namely, the assessment of in-service cable force.  Authors incorporate Hilbert Huang transform to detect the fatigue damage in stay cables using time varying cable tension and its correlation with the  fundamental natural frequency.     The proposed method demonstrates the effectiveness of the proposed technique through field data verification.  This is a novel approach and the results demonstrate the viability of the proposed approach through extensive experimental studies.  I recommend the publicati0on.

Author Response

The authors appreciate the reviewer’s careful comments very much.The authors will continue to work hard to contribute to the intelligent management of bridges.

Reviewer 3 Report

This research focuses on dynamic estimation of cable forces by VMD and HHT. The writing is well-organized and easy to understand. However, there may be important issues related to the interpretations of the test results and applicability of the algorithm considering the objective of the research. 1. Section 1 pp.2 - 3. The definition of variables such as F, L, fn and n are repeated for eq. 1 - 7. These explanations may be omitted or use different variables F1, F2 etc. 2. In the research, the modeling of cables and estimation of natural frequency are different topics. In the proposed method, eventually eq. 2 is adopted. I think it is not necessary to discuss the details of eq. 1 - 7 and simply state eq. 2 is utilized. 3. Section 2 pp. 3 - 4 l. 74 - l. 77. Please state the originality of the research in terms of VMD. VMD is common in signal processing and health monitoring of e.g. machines. If there is no application on stay cables then please discuss the previous research of that field. 4. What is the difference of scale model and full-scale model ? It seems only full-scale model is enough to validate the method. 5. Section 3 p. 9 Fig. 6 (c) (d) and p. 11 Fig. 12. There are some time delays to converge to true cable forces in Fig. 6 (c) (d) around 60 seconds, about 10s - 20s. However, in Fig. 12, the estimation abruptly jumped to true values. Considering the objective of the research, the algorithm should follow the change of the forces quickly to evaluate e.g. fatigue accurately. 6. Is there any true values of forces in the case of real bridge experiment ? 7. In the case of real bridge experiment, the forces at each condition 1 - 3 can be estimated even by previous method. I imagine the change of forces in each condition is the target of the research. Please discuss how the results validate the proposed algorithm.

Author Response

Response to Reviewer 3 Comments

Dear Editors and Reviewers:

Thank you for the valuable and helpful comments for revising and improving the quality of our paper. We have studied the comments carefully and have made the corresponding correction. Revised portion were marked in blue in the paper. The responses to the reviewer’s comments are as following:

Point 1: Section 1 pp.2 - 3. The definition of variables such as F, L, fn and n are repeated for eq. 1 - 7. These explanations may be omitted or use different variables F1, F2 etc. 

Response 1:

The author would like to thank you for your suggestions.

The definitions of repeated variables have been deleted.

Point 2: In the research, the modeling of cables and estimation of natural frequency are different topics. In the proposed method, eventually eq. 2 is adopted. I think it is not necessary to discuss the details of eq. 1 - 7 and simply state eq. 2 is utilized.

Response 2:

The authors appreciate the reviewer’s careful comments very much.

This part is the part of literature review and the authors wants to introduce this part in detail when writing this paper. The unnecessary content has been deleted.

Point 3: Section 2 pp. 3 - 4 l. 74 - l. 77. Please state the originality of the research in terms of VMD. VMD is common in signal processing and health monitoring of e.g. machines. If there is no application on stay cables then please discuss the previous research of that field.

Response 3:

The author would like to thank you for your suggestions.

The innovation of our proposed method lies in the combination of VMD and HHT

The classic frequency-based methods can provide average cable forces while the combination of VMD and HHT can provide the instance cable force, there is little application of stay cables in the research of this filed.

Point 4: What is the difference of scale model and full-scale model ? It seems only full-scale model is enough to validate the method.

Response 4:

The authors appreciate the reviewer’s careful comments very much.

The scales experiment is more easier to conduct in the authors’ lab, the algorithm can be validated quickly. While the full-scale experiment is labour-consuming, more experiment setup works need to be conducted to validate the method in this paper, the full-scale experiment use the real cable that used on the bridges. Both scaled and full-scale experiments are used to verify the algorithm proposed in this paper. The difference between these two models is that a single strand of steel wire was used in the scaled model to simulate the vibration of the cable. Both experiments can prove the accuracy of the algorithm, as shown in Figure 6 and Figure 12.

Point 5: Section 3 p. 9 Fig. 6 (c) (d) and p. 11 Fig. 12. There are some time delays to converge to true cable forces in Fig. 6 (c) (d) around 60 seconds, about 10s - 20s. However, in Fig. 12, the estimation abruptly jumped to true values. Considering the objective of the research, the algorithm should follow the change of the forces quickly to evaluate e.g. fatigue accurately.

Response 5:

The authors appreciate the reviewer’s careful comments very much. The time-varying force should be quickly changed when the true cable force was changed. The adjustment of the true force mainly depends on jack. The fundamental frequency may not be change quickly due to the weight of sensor and cable wire. There may be some time delays when the sensor get the adjusted frequency. Better results can be seen in the full-scale experiments. The cable was the full scale one used on the real bridge, the sensor weight has little influence on the whole cable. There is fewer time delay which can be seen in Figure 12

Point 6: Is there any true values of forces in the case of real bridge experiment ?

Response 6:

The author thinks your questions are very good. Thank you very much.

The forces of real bridges can be measured in many ways. The most accurate method is installing the tension sensor before the cables were installed on the bridge. However, few bridges can be designed considering this aspect. The real bridge in this paper was not install the tension sensor during the initial construction process, the true values of forces are collected by other measurements. These values of forces in the case of real bridge experiment can be found in Figure 17.

Point 7: In the case of real bridge experiment, the forces at each condition 1 - 3 can be estimated even by previous method. I imagine the change of forces in each condition is the target of the research. Please discuss how the results validate the proposed algorithm.

Response 7:

The authors appreciate the reviewer’s careful comments very much.

Through the previous scaled and full-scale experiments, it has been shown that this method can identify the change of cable force. At the same time, through this on-site bridge experiment, it can also be recognized that the cable force before and during the repair are different from those after the repair, which can reflect the changes of actual cable force. However, because the actual bridge structure is very complicated, the actual effect of this method needs to be tested on multiple bridges and compared with the actual bridge cable force measured accurately to obtain better results.

Reviewer 4 Report

I congratulate you for your work and for the important step in this field. It is a narrow but hours of great importance for the engineers of bridges.

Author Response

(The authors gave the same response as above.)

Round 2

Reviewer 1 Report

Dear Authors,

thank you for your answers and please find below my comments. The comment numbers refer to my 1st review.

STAY CABLE EXPERIMENTS

2.

Please indicate these boundary conditions in your manuscript.

3.

Please note that for your lab application (>1 Hz) you could have used a much smaller and lighter accelerometer. Considering the mass of the accelerometer that yous have used - its influence on results is significant.

8.

In case of a cable, this does not seem logical. Please elaborate in text.

14.

Please indicate your response in text.

REAL BRIDGE EXPERIMENT

2. Please indicate your response in text.

6. Please indicate your response in text.

7. Please indicate your response in text.

11. Please indicate your response in text.

Author Response

Response to Reviewer 1 Comments

Dear Editors and Reviewers:

Thank you for the valuable comments for the second revision. The responses to the reviewer’s comments are as following:

STAY CABLE EXPERIMENTS

  1. Please indicate these boundary conditions in your manuscript.

Response:

The authors are very grateful for your comment. The explanation of these boundary conditions has been added in the revised manuscript in L. 91-93.

  1. Please note that for your lab application (>1 Hz) you could have used a much smaller and lighter accelerometer. Considering the mass of the accelerometer that yous have used - its influence on results is significant.

Response:

Due to the limited sensor devices in the lab, the commercial acceleraometer which can provide reliable data was chosen to validate the methodology of proposed methods in scaled and full experiments. The influence may be seen in Fig 6 when the force change delay happened. Full scale experiment showed little influence on the results considering the mass of accelerometer.

  1. In case of a cable, this does not seem logical. Please elaborate in text.

Response:

Thanks for your suggestion. The damaged beam may have higher stiffness values than the undamaged beam before some on-site experiments conducted by the authors. As the reviewer mentioned, the frequencies should be higher with higher cable force. Then the authors checked the original data and manuscript, typo errors were found in the Table 1: the condition 1 should be 6.0 kN while condition 2 should be 5.0 kN. Deeply sorry for this mistake in the previous manuscript.

  1. Please indicate your response in text.

Response:

The explanation of the determination of low-frequencies have been added in the revised manuscript in L.171-175.

REAL BRIDGE EXPERIMENT

  1. Please indicate your response in text.

Response:

The explanation has been added in the revised manuscript in L.232-234.

  1. Please indicate your response in text.

Response:

The explanation of time period selection has been added in the revised manuscript in L.239-240.

  1. Please indicate your response in text.

Response:

The explanation of time period selection has been added in the revised manuscript in L.247

  1. Please indicate your response in text.

The explanation of on-site bridge cable force has been added in the revised manuscript in L.263-264.

Reviewer 3 Report

The reviewer understands most of the points the author answered. Following 2 points are the remaining questions to be addressed. Q. 3 - 5 are some minor comments on the paper. The reviewer would also like the author to improve the readability of the paper. 1. About scale and full-scale models, I understand the scale model was easier to implement. However, loading condition seems similar, and full-scale model is close to a real stay-cable. You can point out the objective of the scale model and indicate the assumed cable model is valid in the both two cases in the paper. 2. I understand the time delay of estimated force in the scale model is due to the weight of a sensor. The reader may be confused because the full-scale model seems more difficult but more accurately estimated than the scale model. The author can discuss this point in the paper. 3. You can emphasize the difference of Eq. 1, 2, 3, 7 (e.g. added term of eq.2 in right hand side). 4. Eq. 3 -7 may be used to derive the formula of F. 5. It seems only the first mode is used to estimate the force. Second and third mode may also be utilized to increase the accuracy.

Author Response

Response to Reviewer 3 Comments

Dear Editors and Reviewers:

Thank you for the helpful comments for revising. The responses to the reviewer’s comments are as following:

Point 1: About scale and full-scale models, I understand the scale model was easier to implement. However, loading condition seems similar, and full-scale model is close to a real stay-cable. You can point out the objective of the scale model and indicate the assumed cable model is valid in the both two cases in the paper.

Response 1:

The author would like to thank you for your suggestions. The full-scale model was difficult to conduct because it is not easy to find a full-scale cable in the experiment. Considering the large size of real cable, it is much easy to use the scaled model to validate proposed method firstly, then a real cable experiment with same load condition (but not the same load values) can be used for further validation. 

The explanation of scaled and full-scale model experiment has been added in Section 3, L.81-88.

Point 2: I understand the time delay of estimated force in the scale model is due to the weight of a sensor. The reader may be confused because the full-scale model seems more difficult but more accurately estimated than the scale model. The author can discuss this point in the paper.

Response 2:

The authors appreciate the reviewer’s careful comments about the discussion of model experiment. The full-scale model is difficult to install but easy to control the load condition, and the mass of sensors can be omitted due to the large size and weight of real cable. That is why the full-scale model experiment results were better than scaled model.

The discussion of this phenomenon has been added in Section 3.3.

Point 3: You can emphasize the difference of Eq. 1, 2, 3, 7 (e.g. added term of eq.2 in right hand side).

Response 3:

The author would like to thank you for your suggestions. Four categories including tense string model theory, simple-supported beam model theory, fixed-supported beam theory, and complex boundary model are introduced and the Equations 1,2,3,7 are the specific explanation. The difference of each equation has been introduced above each equation.

Eq. 1,2,3,7 have been revised by the right hand side.

Point 4: Eq. 3 -7 may be used to derive the formula of F.

Response 4:

The authors appreciate the reviewer’s careful comments very much. As the manuscript mentioned, this model was designed for the fixed-supported beam. Due to the clamped constraint, the displacement and rotation angle at both ends is 0. The cable force can be calculated by the Eq.3-7

Point 5: It seems only the first mode is used to estimate the force. Second and third mode may also be utilized to increase the accuracy.

Response 5:

The authors appreciate the reviewer’s kindly suggestion. This paper was the initial trial to measure the cable force using VMD and frequency-based methods. The first mode was used to estimate the force. From the Fig 5 we can know, it is obvious that the first component (IMF1) was fuller than the other two124components (IMF2 and IMF3), and the value also showed a progressively decreasing form on the order of magnitude, indicating that IMF1 was a relatively major component.

The combination of second and third mode will be considered in the further study.
